High-resolution density assessment assisted by deep learning of Dendrophyllia cornigera (Lamarck, 1816) and Phakellia ventilabrum (Linnaeus, 1767) in rocky circalittoral shelf of Bay of Biscay

Gayá-Vilar Alberto 1
Cobo Adolfo 2
Abad-Uribarren Alberto 3
http://orcid.org/0000-0002-4626-3194 Rodríguez Augusto 3
Sierra Sergio 4
Clemente Sabrina 1
Prado Elena 3 elena.prado@ieo.csic.es
1 Department of Animal Biology, Soil and Geology, University of La Laguna , San Cristóbal de La Laguna, Santa Cruz de Tenerife , Spain
2 Photonics Engineering Group, University of Cantabria , Santander, Cantabria , Spain
3 Santander Oceanographic Center, Spanish Institute of Oceanography (IEO-CSIC) , Santander, Cantabria , Spain
4 Complutum Geographic Information Technologies (COMPLUTIG) , Alcalá de Henares, Madrid , Spain
Vidjak Olja
Electronic publication date: 2024 Mar 7
Publication date: 2024
Volume: 12
Electronic Location ID: e17080
Received 2023 Nov 6; Accepted 2024 Feb 19
Copyright: © 2024 Gayá-Vilar et al.
Copyright year: 2024
Copyright holder: Gayá-Vilar et al.
License: This is an open access article distributed under the terms of the Creative Commons Attribution License, which permits unrestricted use, distribution, reproduction and adaptation in any medium and for any purpose provided that it is properly attributed. For attribution, the original author(s), title, publication source (PeerJ) and either DOI or URL of the article must be cited.
License URL: https://creativecommons.org/licenses/by/4.0/

Keywords: Artificial intelligence, Vulnerable marine ecosystem, Habitat mapping, Object detection model, Natura 2000 network

Funding: LIFE IP INTEMARES Project Biodiversity Foundation of the Ministry for the Ecological Transition and the Demographic Challenge European Union’s LIFE Program LIFE 15 IPE ES 012 This research has been carried out in the scope of the LIFE IP INTEMARES project, coordinated by the Biodiversity Foundation of the Ministry for the Ecological Transition and the Demographic Challenge. This research was funded by the European Union’s LIFE program (LIFE 15 IPE ES 012). There was no additional external funding received for this study. Data collection. ROV images.

==============================
This study presents a novel approach to high-resolution density distribution mapping of two key species of the 1170 “Reefs” habitat, Dendrophyllia cornigera and Phakellia ventilabrum, in the Bay of Biscay using deep learning models. The main objective of this study was to establish a pipeline based on deep learning models to extract species density data from raw images obtained by a remotely operated towed vehicle (ROTV). Different object detection models were evaluated and compared in various shelf zones at the head of submarine canyon systems using metrics such as precision, recall, and F1 score. The best-performing model, YOLOv8, was selected for generating density maps of the two species at a high spatial resolution. The study also generated synthetic images to augment the training data and assess the generalization capacity of the models. The proposed approach provides a cost-effective and non-invasive method for monitoring and assessing the status of these important reef-building species and their habitats. The results have important implications for the management and protection of the 1170 habitat in Spain and other marine ecosystems worldwide. These results highlight the potential of deep learning to improve efficiency and accuracy in monitoring vulnerable marine ecosystems, allowing informed decisions to be made that can have a positive impact on marine conservation.

Introduction

The Habitats Directive (Directive 92/43/EEC) establishes the “Natura 2000” network, a network of European sites which aims to maintain or, if possible, re-establish a favorable conservation status for certain types of natural habitats and certain animal and plant species. The marine Natura 2000 network is an integral part of the European ecological network Natura 2000, and constitutes the application of the Habitats Directive and the Birds Directive (Directive 2009/147/EC) in the marine environment, considered the two most important legislative tools for the conservation of biodiversity in Europe. The Natura 2000 Network is composed of Sites of Community Importance (SCI), which eventually become Special Areas of Conservation (SAC), and Special Protection Areas for Birds (SPA).

The Habitats Directive (92/43/EEC) lists different types of marine habitats that are important for the community and need to be conserved. To do this, Special Areas of Conservation (SACs) are designated. One of the habitats listed in Annex I of the Habitats Directive is Habitat 1170, which refers to Reefs. Reefs in the sense of the Directive are considered to be all those compact hard substrates that outcrop on the seabed in the sublittoral (submerged) or littoral (intertidal) zone, whether of biogenic or geological origin.

In Spain, the Habitat 1170 Reefs extends along the entire coastline and marine waters, from coastal areas to the deep seabed, occupying extensive regions. In this diverse array of Habitat 1170 typologies, our focus narrows to two rocky outcrops within the Cantabrian Sea’s circalittoral shelf. These outcrops are categorized as vulnerable marine ecosystems (VMEs) due to their importance as biodiversity hotspots and ecosystem functioning in the deep sea (FAO, 2009). Circalittoral rocky substrates, located within the phytal system below the maximum distribution level of marine phanerogams and photophilic algae, and extending to the scyaphilic algae’s maximum depth, are characterized by low light levels and relatively stable hydrodynamic conditions compared to shallower regions. The depth at which the circalittoral zone begins depends directly on the amount of light penetrating the seafloor. Animal species predominantly dominate most circalittoral rocky substrates due to the diminished light conditions. The number of species living on these seabeds can be highly variable, influenced by geographical factors, seabed geomorphology, and various environmental elements (Dominguez-Carrió et al., 2022).

Within the Cantabrian circalittoral rocky platform, communities consist mainly of numerous sponge and coralligenous species, which provide three-dimensional structure to these habitats, classifying them under Habitat 1170 Reefs. However, despite their importance as structuring species, their small size and highly fractionated distribution of this organism on the seabed pose significant challenges for mapping. Simultaneously, monitoring these species and tracking community distribution across time and space is imperative for habitat protection. The use of remotely operated vehicles (ROV’s) imagery has emerged as a valuable tool to address this challenge.

Underwater vehicles generate a large amount of in situ, non-destructive, representative, and potentially repeatable samples. This allows not only a complete characterization of benthic diversity but could also lay the groundwork for long-term monitoring initiatives (Dominguez-Carrió et al., 2022). However, processing this information has encountered bottlenecks, primarily attributed to the time-consuming, labor-intensive and costly nature of annotating visual data (Weinstein, 2018). In addition, deep ecosystems present complex environments characterized by unbalanced light conditions, low contrast, and the presence of occlusion and organisms camouflage. Under these circumstances, objects captured by the ROV camera become challenging to identify (Song et al., 2022).

To address these problems and obtain quantitative information from underwater images, new automated image analysis tools have emerged. One of the most promising approaches involves the use of deep learning techniques based on neural networks, a combination of artificial intelligence and computer vision. This approach entails the application of multiple layers of highly interconnected machine learning algorithms to achieve improved results from raw images (Olden, Lawler & Poff, 2008; LeCun, Bengio & Hinton, 2015). These techniques have already achieved formidable results in different marine ecology tasks such as coral classification (Bhandarkar, Kathirvelu & Hopkinson, 2022; Mahmood et al., 2017; Raphael et al., 2020), fish detection and classification (Zhong et al., 2022; Siddiqui et al., 2018; Knausgård et al., 2022), and identification of diverse benthic fauna (Abad-Uribarren et al., 2022; Song et al., 2022; Liu & Wang, 2022).

Within the field of deep learning, object detectors can be classified into two categories: two-stage detectors and single-stage detectors. Two-stage detectors exemplified by Faster region-based convolutional neural network (Faster R-CNN) (Ren et al., 2017), first generates a set of region proposals (RPN) before determining the object category and location. In contrast, single-stage detectors, such as YOLO (Redmon et al., 2016), simultaneously identify and locate objects in a single step. These object detection models can be used as tools to automate species identification procedures and generate accurate density maps for ecosystem monitoring (Prado et al., 2020). Surveys employing these models yield comprehensive records of ecosystems and facilitate the identification of trends in habitat health and biodiversity.

Marine ecosystems are subject to numerous threats and impacts, such as climate change, waste pollution, commercial fishing and deep sea mining (Pinheiro et al., 2023). Therefore, it is important to generate detailed mapping of their most vulnerable ecosystems in order to make informed decisions about their management and conservation. Accurate mapping facilitates the identification of critical areas that require protection and the development of effective strategies to mitigate negative impacts on the ecosystem (Rodríguez-Basalo et al., 2022).

In this study, we assess the density of Habitat 1170 structuring species in two circalittoral rocky shelf areas of the Cantabrian Sea. We employ object detection models to automatically identify and label species in underwater images. Initially, we compare object detection models with different neural architectures, to determine the most effective model for generating species density maps from geolocated images obtained along a photo-transect. Subsequently, the model demonstrating the best metrics is employed to establish a pipeline for generating detailed species density maps. Our ultimate goal is to create an initial base map for monitoring ecosystem health, offering a comprehensive geographic description of Habitat 1170 structuring species, and serving as a support tool for decision-making in Natura 2000 network areas. The expected results of this model’s application include the automated generation of density and geographic presence data for benthic species Dendrophyllia cornigera and Phakellia ventilabrum, with results presented through species density maps.

Materials and Methods

Study area

The research was centered on two rocky outcrops situated within the circalittoral shelf of the Aviles submarine Canyons System (ACS) and the Capbreton submarine Canyons System (CCS), both located in the Cantabrian Sea to the south of the Bay of Biscay (Fig. 1). The ACS has been designated as a Site of Community Interest (SCI) and is currently undergoing studies aimed at elevating its status from SCI to Special Area of Conservation (SAC). Likewise, the CCS area is under examination for SCI status, with the intention of integrating it into the marine Natura 2000 Network. These studies are part of the actions carried out in the LIFE IP INTEMARES project (Baena et al., 2021).

Figure 1 Geographical and bathymetric overview of study areas in the Cantabrian Sea.

(A) Location map showing the study areas highlighted by green rectangles, located in the Cantabrian Sea. In addition, detailed representations of the bathymetry of (B) the Aviles submarine canyon system and (C) the Capbreton submarine canyon system are presented, where the ROV transects identified by red dots are highlighted.

In the Bay of Biscay, the continental shelf is generally narrow, a common feature of compressive continental margins (Ercilla et al., 2008). The region is characterized by the presence of rocky outcrops mainly formed due to sedimentary transport mechanisms associated with oceanographic dynamics. These rocky outcrops serve as critical habitats for a diverse array of benthic communities, many of which fall within the 1170 habitat category forming a heterogeneous and complex ecosystem capable of supporting a rich biodiversity. Within these communities, our study focuses on two species selected for their significant role in structuring the 1170 habitat within the rocky circalittoral platform of the Cantabrian Sea (Rodríguez-Basalo et al., 2022). These species are the yellow coral Dendrophyllia cornigera (Lamarck, 1816) and the cup sponge Phakellia ventilabrum (Linnaeus, 1767) (Fig. 2).

Figure 2 Detailed Photographs of Dendrophyllia cornigera and Phakellia ventilabrum.

(A) Dendrophyllia cornigera and (B) Phakellia ventilabrum.

ROTV underwater imagery

High-resolution underwater images obtained using the remotely operated towed vehicle (ROTV) Politolana (Sánchez & Rodríguez, 2013) were employed. The ROTV Politolana, designed by the Santander Oceanographic Center of the Spanish Institute of Oceanography (IEO-CSIC), has the capability to descend to a maximum depth of 2,000 m. For seabed exploration, the ROTV Politolana uses photogrammetric methods and is equipped with a high-resolution camera, bidirectional telemetry and an acoustic positioning system. Additionally, the vehicle is equipped with four laser pointers coupled at a precise distance of 25 cm from each other. This configuration allows precise measurements and detailed data to be obtained during each deployment. Furthermore, the vehicle acquires high-definition images and videos synchronized with environmental data, ensuring the acquisition of comprehensive datasets during each dive.

In total, 20 transects were conducted for this study, with an average length of 410 m per transect. In the ACS, images were acquired during the INTEMARES A4 Avilés oceanographic campaign (2017). In contrast, in the CCS, photographic transects were carried out during the INTEMARES-Capbreton 0619 and 0620 campaigns (2019 and 2020). These transects, both in ACS and CCS, were carried out in a depth range between 90 and 300 m.

The Politolana ROTV captures photographs at time intervals ranging from 0.5 and 20 s, depending on the chosen sampling configuration. This approach provided representative data of the habitat and benthic communities to be characterized. These high resolution images provided comprehensive views of the seafloor in Fig. 3. In total, 5,012 images were contributed from both ACS and CCS for this study.

Figure 3 ROTV captured images of marine species in Capbreton and Aviles Canyon Systems.

(A) Capbreton Canyon System, showing Dendrophyllia cornigera (Dc), Phakellia ventilabrum (Pv), encrusting sponges (Ei), other Porifera organisms (Po), Viminella flagellum coral (Vf), and Filograna cf implexa serpulid (Fi). (B) Aviles Canyon System, displaying Dc, Ei, Po, Pv, Artemisina transiens (At), Leptometra celtica (Lc), Parastichopus regalis (Pr), and Gracilechinus acutus urchin (Ga). Lasers are visible in the center of both images.

Image preprocessing, algorithm training and validation

To evaluate the model’s generalization capabilities, 60 images from a pool of 300 obtained from the ACS were randomly selected as the validation set. The remaining 240 images were used as training images in a ratio of 8 to 2. To prevent overfitting, a careful selection process ensured no repetition of images between the training and validation sets. Each image was uniquely identified, and during validation set creation, checks were implemented to avoid duplications. This approach safeguards model integrity by ensuring independent representation in both training and validation sets.

For image annotation, we utilized the Supervisely image data annotation software (https://supervise.ly/), enabling the creation of bounding boxes around the target species D. cornigera and P. ventilabrum. Our labeling approach ensured that each annotation encompassed the entire individual while minimizing background area (Fig. 4). The training data set received meticulous attention, with 527 and 1,045 annotations performed for the D. cornigera and P. ventilabrum classes. Annotations were carried out by trained expert scientists. To balance class distribution, the Supervisely flying object function (https://github.com/supervisely-ecosystem/flying-objects) was used. This function generates synthetic data for object detection tasks by annotating specimens from both classes as masks, applying magnifications to the objects, and distributing them on different selected backgrounds. With the creation of 20 synthetic images, class balance was achieved with 2,330 and 2,605 annotations for the classes D. cornigera and P. ventilabrum, respectively (Fig. 4).

Figure 4 Synthetic image generation of D. cornigera and P. ventilabrum using supervisely.

(A) Specimens of both D. cornigera and P. ventilabrum. (B) Specimens of D. cornigera only. (C) Specimens of P. ventilabrum only. Each image, enhanced by the flying object function for magnification and sample size increase, contains an average of 120 annotations.

For image evaluation, we employed a k-fold cross-validation method to estimate model prediction uncertainty. The dataset was randomly partitioned into five subsets, allowing us to compare training and testing sets. This facilitates obtaining realistic estimates of predictive performance on previously unseen data and evaluating uncertainty surrounding the fitted functions. This cross-validation process was applied to the 240 annotated images used for training. Following each of the five training sets, we introduced synthetic images with augmentations to analyze how models respond to data augmentation. After completing the cross-validation, we assess our models using the validation images, exploring their performance both with and without the inclusion of augmented images in the training set.

In addition, to assess the model’s capacity for generalization across diverse environmental conditions, 60 images were randomly selected from the CCS area, a region where the two target species are exposed to different environmental pressures. These images served as validation data to determine the model’s proficiency in accurately detecting the target species.

All object detection models were based on the same pre-training weights from the COCO (Common Objects in Context) dataset, which is a widely used dataset in computer vision research. The models were trained on selected ACS images, both with and without data augmentation. The training spanned 200 epochs, allowing the models to continually improve their accuracy and performance. During each epoch, the models processed the entire training data set, iteratively adjusting their parameters to minimize prediction error. This allowed the models to continuously improve their ability to accurately detect the target species.

Automatic species labeling of underwater images using deep learning

For automatic annotation of the image set, a deep learning based framework was developed, considering three different deep neural network architectures. A neural network consists of input layers that receive the data, a processing core with hidden layers, and output layers that provide the model output. The term “deep” refers to the number of hidden layers in the neural network structure. A neural network is trained to produce the desired output by adjusting its internal parameters, called weights, based on the error between the model output and the correct response. This adjustment is performed by a process called gradient descent (Schmidhuber, 2014).

This study compared three object detection models with different neural architectures: YOLOv7 and YOLOv8 (Bochkovskiy, Wang & Liao, 2022; Terven & Cordova-Esparza, 2023), both single-stage models, and Faster R-CNN (Ren et al., 2017), a two-stage model. YOLO single-stage models have been shown to have advantages compared to Faster R-CNN (Abdulghani & Menekşe Dalveren, 2022; Maity, Banerjee & Sinha Chaudhuri, 2021; Li et al., 2022), so the latter two model versions were included in the comparison.

YOLOv7 and YOLOv8 are real-time object detection models that employ convolutional neural networks (CNNs) to efficiently identify and localize objects in images. YOLOv7 uses the ELAN architecture, which improves the learning and convergence capabilities of deep networks. On the other hand, YOLOv8 integrates advances in deep learning and computer vision, including attention structures and dilation convolution blocks, resulting in improved speed and accuracy in object detection. In this study, the YOLOv8x model was used for YOLOv8 training, while the YOLOv7-E6E model was used for YOLOv7 training.

Faster R-CNN, a two-stage object detection model, integrates a CNN for features extraction and a Region Proposal Network (RPN) to generate high-quality proposals. The RPN predicts the boundaries and objectivity scores at each image position and is trained end-to-end. These proposals are then used by Faster R-CNN in the object detection and classification stage (Ren et al., 2017). In this study, the X-101-32x8d model was used for Faster R-CNN training.

The YOLOv8x, YOLOv7-E6E and Faster R-CNN X-101-32x8d versions were chosen for this study because of their object detection performance. According to the data and the information available in the GitHub repositories, these versions present high performance in terms of accuracy and speed in object detection.

For training the object detection models, we used Google Colab Pro, a platform that provided us access to the NVIDIA A100-SXM GPU. This high-performance GPU enabled efficient processing of large datasets and expedited model training. Furthermore, Google Colab facilitated seamless code sharing and result dissemination among team members.

Object detection model selection

In order to select the best-performing model for the task of detecting D. cornigera and P. ventilabrum, we utilized three widely recognized metrics for object detection: precision, recall, and F1 score. These metrics underwent a comprehensive comparison and analysis across the YOLOv8, YOLOv7, and Faster R-CNN object detection models.

Precision measures the accuracy of the model predictions, representing the percentage of predictions that are correct.

Recall (or the sensitivity of a classifier) evaluates how effectively the model identifies all positive instances, quantifying the number of actual positives correctly labeled as positives.

The F1 score serves as an index that evaluates the balance between precision and recall, a widely used metric in deep learning for comparing the performance of two models on the same task. The calculations for precision, recall and F1 are described by the following Eqs. (1)–(3) (Van Rijsbergen, 1974):

(1) precision=TruepositivesTruepositives+Falsepositives

(2) recall=TruepositivesTruepositives+Falsenegatives

(3) F1=2⋅precision⋅recallprecision+recall

To determine the confidence threshold for presentation of our results and model selection, we employed the receiver operating characteristic (ROC) curve method to assess the predictive accuracy of the models, calculating the area under the ROC Curve (AUC) as a measure. Additionally, we compared the precision and recall curves of each model to help establish the confidence threshold that would represent our data. Given the abundance of our target species in the environment, we selected the threshold by prioritizing precision, aiming to avoid overestimating the populations of our target species. This ensures that, when generating maps, populations are not overestimated. Model evaluation was based on Intersection over Union (IoU) with a value of 0.5, quantifying the overlap between detection and the real object (Fig. 5). An IoU of 0.5 indicates that the detection covers 50.0% of the actual object’s area (Nowozin, 2014).

Figure 5 Intersection over union (IoU) calculation for D. cornigera detection.

Orange (A) shows the bounding box of the annotation and blue (B) shows the bounding box of the inference, as well as the center of each box and the distance between both points. The IoU is calculated as the intersection between the two boxes divided by their junction.

Both ACS training datasets, with and without augmentations, were used for evaluating the models on the ACS and CCS validation datasets. The addition of the CCS dataset was crucial for assessing the model’s ability to generalize to new images of other canyon systems subjected to different pressures and environmental conditions. A comprehensive evaluation of the models was conducted under different scenarios and conditions, ensuring the acquisition of accurate and reliable results for object detection in underwater systems. The main aim is to find the model with the best results, laying the groundwork for accurate maps based on species distribution.

Pipeline for species density map generation

To streamline the process of generating species density maps from raw transect images, a pipeline was implemented in Google Colab Pro (Fig. 6). The images were synchronized with the ROTV telemetry data, which provided information on the depth, coordinates, and height of the ROTV relative to the seafloor for each transect image. In parallel, using ImageJ software (version 1.53o), the distance between the ROTV laser pointer marks on 50 images was manually measured to obtain the area covered by each image based on its resolution. With this data, a simple regression model was trained using supervised machine learning techniques, specifically employing linear regression, to establish a relationship between the area and the height of the ROTV relative to the seafloor. The model predictions were used to calculate the area of the rest of the images.

Figure 6 Automated species density map generation pipeline in google colab.

Generation of species density maps from raw transect images and ROTV telemetry data.

Simultaneously, predictions were carried out using the YOLOv8 model, previously trained with our data (YOLOv8-SCS), to analyze the images captured in the transects. Notably, in the comparison of object detection models, YOLOv8 obtained the best metrics. Therefore, we set these parameters to perform the predictions and generate the inferences in the workflow. These results were integrated with the area and coordinates of each image in order to calculate the number of individuals per square meter in each image.

Finally, QGIS software (version 3.22; QGIS, London, UK) was used to create maps based on the species density data obtained from each transect image point. We applied a symbology scheme based on graduated symbol sizes to represent density categories (Schmidt et al., 2022). The density data series was classified into intervals according to their values, and each interval was assigned a corresponding symbol size, with larger sizes indicating higher densities. This approach facilitated the clear visualization of areas with the highest species density on the generated maps.

Results

Comparison of object detection models

In the evaluation of object detection models, we compared one-stage models YOLOv7 and YOLOv8 with the two-stage model Faster R-CNN. Our assessment included both models trained with and without data augmentation, and the results are summarized in Table 1. Notably, data augmentation improved the performance of all three models, with the one-stage models demonstrating more substantial enhancements for D. cornigera. YOLOv7 achieved a recall rate of 7.2% and an F1 score of 5.8%, while YOLOv8 showed a recall improvement of 4.7%. For P. ventilabrum species, the Faster R-CNN model showed the most significant improvement with an 8.7% increase in recall and a 3.9% in F1 score. The YOLOv8 model also showed a 4.3% improvement in precision for P. ventilabrum.

Table 1 Impact of data augmentation on object detection model performance for D. cornigera and P. ventilabrum.

Species	Model	Precision (%)	Recall (%)	F1 (%)	
D. cornigera	YOLOv8	0.8	3.7	1.3	
D. cornigera	YOLOv7	4.3	0.6	1.1	
D. cornigera	Faster R-CNN	2.5	8.7	3.9	
P. ventilabrum	YOLOv8	4.9	7.2	5.8	
P. ventilabrum	YOLOv7	1.7	4.7	2.5	
P. ventilabrum	Faster R-CNN	0.2	1.7	0.4	
Note:

Difference in performance of YOLOv8, YOLOv7 and Faster R-CNN models in terms of precision, recall and F1 for D. cornigera and P. ventilabrum species with and without data augmentation. Values represent the difference between model performance with data augmentation minus model performance without data augmentation.

When examining the results of k-fold cross-validation (Fig. 7) and validation in ACS and CCS (Table 2), the consistency in the models’ performance is confirmed, supporting the selection of YOLOv8 as the standout model in terms of precision and F1 score. This model demonstrates a notable ability to generalize to new data, essential for its implementation in diverse underwater environments.

Figure 7 Cross-validated species detection performance.

Comparison of K-fold cross-validation metrics for detecting P. ventilabrum and D. cornigera species across YOLOv8, YOLOv7, and Faster R-CNN models. The evaluation includes precision, recall, and F1 score metrics, with analysis conducted on augmented.

Table 2 Species detection metrics comparison.

Canyon	Species	Model	Precision (%)	Recall (%)	F1-score (%)	
Aviles	D. cornigera	YOLOv8	88.9	62.9	73.7	
Aviles	D. cornigera	YOLOv7	92.4	91.0	91.7	
Aviles	D. cornigera	Faster R-CNN	87.1	84.5	85.8	
Aviles	P. ventilabrum	YOLOv8	91.4	75.5	82.7	
Aviles	P. ventilabrum	YOLOv7	91.2	92.1	91.6	
Aviles	P. ventilabrum	Faster R-CNN	89.9	90.1	90.0	
Capbreton	D. cornigera	YOLOv8	65.0	54.1	59.1	
Capbreton	D. cornigera	YOLOv7	92.5	79.0	85.2	
Capbreton	D. cornigera	Faster R-CNN	77.6	72.3	74.9	
Capbreton	P. ventilabrum	YOLOv8	71.7	62.0	66.5	
Capbreton	P. ventilabrum	YOLOv7	88.9	83.7	86.2	
Capbreton	P. ventilabrum	Faster R-CNN	85.8	82.2	84.0	
Note:

Comparison of metrics for detection of P. ventilabrum and D. cornigera species using the YOLOv8, YOLOv7 and Faster R-CNN models in the ACS and CCS. precision, recall and F1 metrics were analyzed using augmented data.

Upon analyzing the results specific to each species and location, distinctive patterns emerge. YOLOv8 outshines the other models for both species, with values exceeding 91.0% in precision, recall, and F1 in ACS, and an F1 score surpassing 85.2% in CCS. Specifically, for D. cornigera in Capbreton, it stands out with a precision of 92.5%. These values indicate high precision and an effective balance between precision and recall for this particular model. Meanwhile, considering YOLOv7 reveals notable challenges, especially in recall, reflecting lower metrics in both datasets. Faster R-CNN, though exhibiting strong performance with F1 values exceeding 80.0% in most metrics, struggled to generalize for D. cornigera in Capbreton.

In terms of consistency between cross-validation and validation results, values in Table 2 mirror similarities in trends, supporting YOLOv8’s generalization capability. Additionally, it is noteworthy that YOLOv8 displays the lowest error deviation in cross-validation (Fig. 7). However, these observations also highlight opportunities to enhance the robustness of YOLOv7 and explore specific adjustments for Faster R-CNN in future investigations.

Figure 8 presents a visual representation of the detection results obtained using the YOLOv8, Faster R-CNN and YOLOv7 models. The models occasionally misclassified certain sponges as P. ventilabrum, especially Faster R-CNN, which showed more false detections, including instances where D. cornigera was misidentified as other sponge species that warrant in our annotation class. As outlined, our classification model focused exclusively on distinguishing between the two specified classes, D. cornigera and P. ventilabrum, without considering additional classes in our annotation. Notably, none of the models detect all D. cornigera specimens in the images, with YOLOv7 showing the highest number of undetected specimens, missing over 60.0% of them. YOLOv8 stands out for its ability to detect smaller D. cornigera specimens at high densities. In a separate set of images, Faster R-CNN misclassifies a complex sponge as both D. cornigera and P. ventilabrum and also inaccurately identifies encrusting sponge specimens as D. cornigera. The YOLO-based models struggle to detect P. ventilabrum specimens accurately.

Figure 8 Detection results comparison of different algorithms for CCS images.

(A and B) Different images. Red circles indicate false detections, while blue circles indicate missed detections. The detection model used is shown in the right frame of each image.

Overall, these results demonstrate that YOLOv8 exhibits superior detection efficiency and counting precision compared to other evaluated models, making it the preferred algorithm for detecting these valuable marine benthic species.

Species density survey

A total of 5,021 transect images were processed through a pipeline designed for automatic detection of target species. These transects covered an area of 5,647.5 m2, with an average area covered of 282.4 m2. As a result, the YOLOv8-SCS model generated 27,668 automatic annotations comprising 6,087 for P. ventilabrum and 21,581 for D. cornigera. This resulted in an average density of 1.0 individuals/m2 for P. ventilabrum and 3.1 individuals/m2 for D. cornigera in the ACS and 1.2 individuals/m2 for P. ventilabrum and 5.0 individuals/m2 for D. cornigera in the CCS.

Species density maps were generated based on this information, allowing for a comparison of the two study areas. The CCS rocky platform exhibited the highest densities for both species, with a maximum density of 60.6 individuals/m2 for D. cornigera and 13.0 individuals/m2 for P. ventilabrum. These peak density observations occurred within the same transect at an average depth of 160.1 m and are illustrated in the species density maps (Fig. 9).

Figure 9 Species density maps for D. cornigera and P. ventilabrum in Aviles and Capbreton Canyon Systems.

(A) Dendrophyllia cornigera and (B) Phakellia ventilabrum. Both species are shown in the Aviles Submarine Canyon system (bottom left) and the Capbreton Canyon System (bottom right). A large-scale situation map at the top depicts the locations of both canyon systems.

The implementation of the YOLOv8-SCS model for automatic annotation has led to significant time savings. While a professional researcher would require approximately 210 h 53 min to annotate all 5,021 images, the YOLOv8-SCS model accomplished the same task in just 2 h 9 min. This represents a time reduction of over 99.0%, highlighting the efficiency and effectiveness of the automatic annotation process.

Discussion

The present study addresses the need to provide efficient solutions for the monitoring, protection and conservation of vulnerable marine ecosystems (VMEs) of the circalittoral shelf by implementing automatic identification algorithms and calculating densities without human intervention. These ecosystems are crucial for maintaining marine biodiversity and play a vital role in the provision of essential ecosystem services (Ríos et al., 2022). Incorporating automatic image analysis techniques represents a significant advancement in the field of benthic community studies (Abad-Uribarren et al., 2022). The tremendous diversity of species present poses a formidable challenge for an exhaustive species analysis approach. Nevertheless, cataloging the species and their sizes provides an invaluable means to analyze community composition. Unfortunately, this aspect is frequently overlooked due to the considerable time it consumes in the evaluation process (Schoening et al., 2012). To this end, we have used deep learning tools to assist in the identification and mapping of these VMEs of rocky circalittoral shelf areas adjacent to the headwaters of two submarine canyon systems, the ACSs and CCSs. The choice of YOLOv8 as the algorithm for this task was influenced by its efficient information processing capabilities and sophisticated architecture that includes advanced loss functions. It is worth noting that various versions of the YOLO algorithms have been adapted to tackle specific challenges of underwater images, such as lack of sharpness, small size, and overlap (Zhang et al., 2022; Xu et al., 2023). The continuous evolution of these algorithms and the release of newer versions present an opportunity to test each version on the same datasets to quantify improvements in results (Zhong et al., 2022). The latest version, YOLOv8, has already shown promise in plant species recognition (Wang et al., 2023), yet its application in marine environments remains largely unexplored. This opens up new research avenues and potential enhancements to our current methodology.

Emerging from this, our study has made significant strides in applying YOLOv8 to marine environments, demonstrating its high efficacy despite the inherent challenges posed by morphological variations among specimens of the same species, such as the diversity in shapes, sizes, and complexity of D. cornigera colonies, and the variability in sizes and forms of P. ventilabrum. Notably, YOLOv8 exhibited minimal variance in errors during training validation, indicating consistent performance, stability, and robustness against overfitting (Fig. 7). In the validation phase, YOLOv8 achieved F1 values exceeding 91.6% for both species in the ACS, highlighting a high level of precision in detection and recovery. Additionally, it successfully detected both species with accuracies exceeding 88.9% in the CCS. These results not only corroborate Li et al.’s (2022) findings, affirming YOLOv8’s suitability for complex conditions and its universality and robustness in object detection amid variability and noise, but also significantly contribute to our understanding of rocky circalittoral platform habitats and the distribution patterns of vulnerable species within Natura 2000 spaces.

The success in species detection is attributed to the prior training of models on large-scale datasets, such as COCO, and the augmentation of data through synthetic images, a beneficial strategy to address class imbalance and enhance metrics. Specifically, a 2.5% improvement in F1 was observed by including synthetic images for the D. cornigera class. However, the generalization capacity of models is crucial to ensure the advancement of Deep Learning applications in ecological studies. Generalization issues are common, where algorithms trained on specific datasets may not perform adequately in new areas or with unconsidered species (Xu & Matzner, 2018). YOLOv8 demonstrated excellent generalization capabilities, showing robust performance in predictions for ACS and CCS validation images. In contrast, Faster R-CNN provided comparable performance but with slightly lower metrics, and YOLOv7 struggled to generalize effectively when transitioning to CCS. Validation with Capbreton images (CCS) was crucial to assess model overfitting. An alternative approach to improve model generalization involves increasing manual annotation efforts, albeit incurring significant personnel and time costs (Weinstein et al., 2022). Although our research focuses on two specific species, it is crucial to highlight the ease with which an effective tool can be developed using a limited number of images. It is essential to note that our study is conducted in a circalittoral reef, where target species are dispersed, unlike tropical coral reefs where they often form complex high-density clusters.

The analysis carried out in this study provides valuable insights into the distribution patterns of two target species within and between our study areas. The species are not uniformly distributed, but seem to present a patchy distribution with higher or lower associated densities depending on the geographic location. This finding aligns with previous research, such as the study by Rodríguez-Basalo and colleagues in 2022 in ACS, which also observed differences in the densities of D. cornígera and P. ventilabrum.

In our study, we found that P. ventilabrum has densities ranging from 45.3 to 173 ind/100 m2, while D. cornígera shows densities ranging from 7.5 to 149.3 ind/100 m2. This complex distribution pattern highlights the need for a comprehensive dataset consisting of multiple spatial images to accurately capture the nuances in species distribution. From an ecological and management perspective, understanding these density differences in our study areas is of great importance. Our work provides a detailed mapping of density variations for these species, serving as a foundation for future research into the underlying causes of these variations. These causes could be related to differences in environmental conditions between the study areas, varying levels of human influence, or a combination of both factors.

Our findings indicate that D. cornigera and P. ventilabrum have higher densities in the CCS compared to the ACS. Furthermore, the ACS displayed a more intricate but less abundant coral community, along with a higher prevalence of encrusting sponges and fewer three-dimensional sponges. These differences did impact the performance of our models in both study systems. However, it is important to note that despite these varying characteristics, our model demonstrated a high degree of accuracy and proficiency in object detection. It is worth emphasizing that the main focus of this work is the development of an automated annotation tool, and we do not provide an exhaustive ecological interpretation of the data generated by this tool. This limitation highlights the need for further research to explore the ecological implications of the observed distribution patterns of these two species in the study areas.

Once YOLOv8 was identified as the most effective model for the detection of the target species, we proceeded to automate the process of obtaining ecological data from raw images. The use of deep learning algorithms allows us to generate accurate and efficient species density maps (Fig. 9). The time required to perform all image labeling using these automatic models is drastically reduced compared to the time required for this same task by experts, who must manually search and label each species present in thousands of images. Therefore, these models provide a great advantage, since they provide valuable information in a minimum amount of time for carrying out subsequent in-depth population and ecological studies, which result in the improvement of management measures applied in the ecosystems studied.

The presence of high densities of species belonging to the 1170 Reef habitat supports the need to establish regulations and sustainable management measures to preserve biodiversity in the ACS and CCS. These ecosystems are being considered for protection and conservation through their designation as SAC and SCI, thanks to the LIFE IP INTEMARES project, which is aligned with the global objective of reaching 30.0% of marine protected areas by 2,030 under the umbrella of the Natura 2000 Network. This study demonstrates the importance of using advanced technologies to comprehensively study these complex and deep ecosystems serving as a basis for the establishment of appropriate legislative measures, supported by rigorous scientific information with a high degree of detail, in order to maintain the balance of the structure of these benthic ecosystems, of the trophic relationships they support, and to ensure the sustainability of the fisheries they support.

The implementation of a pipeline as a management and conservation tool for ACSs and CCSs provides the opportunity to monitor the densities of the shelf area study species. In addition, the pipeline has the potential to evolve and adapt to the needs of future surveys to obtain real-time data, cover a larger number of species, and provide more detailed information on the sizes of detected organisms. In addition, their ability to replicate over time will favor the monitoring of the environmental evolution of these areas or their possible response to the management measures applied. Several studies, such as Zhong et al. (2022), have successfully demonstrated the use of YOLO-based object detection models for real-time identification of marine animals, supporting the validity of the approach. To further enhance the results, transformer-based models like ViT and CLIP (Dosovitskiy et al., 2020; Radford et al., 2021), could also be considered in future research. For the extraction of species measurements there are promising state-of-the-art segmentation models such as the Segment Anything Model (SAM) that would allow us to obtain the area occupied by the species in the image in real time (Kirillov et al., 2023). To avoid biases and enhance generalization, it is crucial to conduct tests in various locations and broaden the detection of species (Li et al., 2022). This will enrich our study and provide more precise details about the structure of local communities. This will also ensure measurement accuracy by establishing a protocol for ROV image capture that minimizes errors and synchronizes ROV data with captured images. This tool would provide a holistic perspective, being able to detect changes in marine life patterns in terms of density, mean sizes and biomasses, in order to properly assess the impact of human activities on these habitats. This would facilitate the adoption of measures to protect and conserve these unique marine ecosystems.

Regarding possible improvements of the present work, it is suggested to use a larger and more diverse set of high quality images to train the model. It is proven that increasing the number of images during training significantly improves the generalizability and accuracy of the object detection model (Eversberg & Lambrecht, 2021; Zoph et al., 2020).

Conclusion

1) The YOLOv8 model was effective for the detection of the yellow coral Dendrophyllia cornigera and the cup sponge Phakellia ventilabrum, two key species of the Cantabrian Sea circalittoral shelf rock.

2) A powerful and accurate tool was developed, within a pipeline, that allows automatic detection of target species from raw transect images of the circalittoral shelf by remotely operated vehicles (ROVs).

3) The results show that all three models (YOLOv7, YOLOv8 and Faster R-CNN) improve their performance when trained with data augmentation and that YOLOv8 is the model that presents the best performance in terms of precision, recall and F1.

4) The implementation of this tool in the shelf area of the Aviles (ACS) and Capbreton Submarine Cannon systems (CCS) allowed monitoring of these vulnerable marine ecosystems, with detailed density maps of target species indicating that the CCS rocky shelf presented the highest densities D. cornigera and P. ventilabrum.

5) The implementation of deep learning based technologies are an efficient and accurate methodology for sampling and monitoring sessile benthic populations. This is essential to support the protection and conservation of biodiversity in these ecosystems.

Supplemental Information

Supplemental Information 1 The Python script detailing the pipeline used for data extraction from images, including steps for area and species count.

All the code for converting RAW images into available data for mapping.

Supplemental Information 2 Raw data from the ROV.

The raw data collected from the remotely operated towed vehicle (ROTV) used in our study. It includes various measurements and observations essential for our deep learning model to perform high-resolution mapping of key reef species in the Aviles and Capbreton Canyon Systems.

Supplemental Information 3 Data from a regression model used to extrapolate area from images.

Supplemental Information 4 KFold Cross-Validation.

The splits of the KFold cross-validation k = 5.

Supplemental Information 5 Final Data for High-Resolution Species Density Mapping.

The final data used for creating high-resolution species density maps in our study. This data, generated by our deep learning model, was used in QGIS to visualize the distribution of key reef species in the Aviles and Capbreton Canyon Systems.

Supplemental Information 6 Example image to run the Pipeline.

The authors would like to thank the crew and scientific team aboard the R/V Ramón Margalef from the Spanish Institute of Oceanography and also appreciate the helpful assistance of the technicians of the ROTV Politolana for their skill in executing the dangerous visual transects very close to the rough bottoms of the study area.

Additional Information and Declarations

Competing Interests

Author Contributions

Data Availability

The authors declare that they have no competing interests.

Alberto Gayá-Vilar conceived and designed the experiments, performed the experiments, analyzed the data, prepared figures and/or tables, authored or reviewed drafts of the article, and approved the final draft.

Adolfo Cobo conceived and designed the experiments, analyzed the data, authored or reviewed drafts of the article, and approved the final draft.

Alberto Abad-Uribarren conceived and designed the experiments, authored or reviewed drafts of the article, and approved the final draft.

Augusto Rodríguez conceived and designed the experiments, prepared figures and/or tables, authored or reviewed drafts of the article, and approved the final draft.

Sergio Sierra conceived and designed the experiments, analyzed the data, authored or reviewed drafts of the article, and approved the final draft.

Sabrina Clemente performed the experiments, authored or reviewed drafts of the article, and approved the final draft.

Elena Prado conceived and designed the experiments, analyzed the data, authored or reviewed drafts of the article, and approved the final draft.

The following information was supplied regarding data availability:

The code and raw data are available in the Supplemental Files.

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
