# Peer review of "High-resolution density assessment assisted by deep learning of Dendrophyllia cornigera (Lamarck, 1816) and Phakellia ventilabrum (Linnaeus, 1767) in rocky circalittoral shelf of Bay of Biscay"

_PeerJ, doi:10.7717/peerj.17080_

## Round 0.1 · original submission · Minor Revisions

We have received two reviews of your paper, both of which recognise the merits of your work. I consider both to be helpful and aim to improve the content. Therefore, please address the suggested changes and comment on the issues raised by responding point by point. Please also correct some minor errors marked in the text of the attached pdf file (editorial corrections).

Reviewer 1 ·

Basic reporting

On the whole, the paper is very readable, with clear English used throughout.

Introduction and context clear. Literature review contains a good mix of standard citations for the field and recent citations which will make it a useful reference point for readers interested in the subject.

The structure follows the norms for the discipline.

Most of the figures and tables are clearly presented and captioned (see later comments). Figure 7 needs reconsidering as it attempts to compare 3 algorithms, 2 confidence thresholds, precision-recall-F-score, 2 species and 2 sites. These scores are based on a single run, rather than means of cross-validation (see later comment).

Raw data (images) are not supplied, but the supporting information identifies the images used. Authors should make it clear how to obtain the primary data to replicate the study.

Minor typos:
63 - 2 periods
79 - typo "generates"
147 - 2 periods
271-273 - Datasets are not referred to by their abbreviations
275 - typo ".A"
280 - typo ", A"
548 - Author missing
570 - Author missing
572 - Author missing

Experimental design

Overall research is within scope for the journal.

The contribution of comparing 3 algorithms on 2 datasets is useful and will be of interest to practitioners in this area.

The research focuses on identifying 2 species in a relatively sparse environment, mostly avoiding issues of overlapping individuals (based on the images in the paper). Both species appear to be morphologically stable in the chosen environments (again, based on the images in the paper). These factors make the task of automatically identifying the species easier than, say, a diverse tropical coral reef with high benthic cover with morphologically variable species. It would be important for the authors to contextualise their findings within this scope.

Given the above, using bounding boxes rather than polygon or pixel-level annotation seems reasonable; however, it does reduce the usefulness of the findings. Most recent work on automatic benthic annotation uses pixel masks or polygon shapes rather than bounding boxes, making the algorithm processing more difficult but the end results for IoU and F-scores much more accurate and meaningful.

The authors state the algorithm mistook "D. cornigera for other sponge species" (line 335), suggesting that a multi-class classifier was being used but not reported on? This needs clarifying.

The image processing method does not seem to follow standard cross-validation protocol (whereby the dataset is randomly divided into, say, 10 sets of 80:10:10 (training, testing, validating)). Using 10-fold cross-validation reduces the effect of selection bias (as you can report mean F and standard deviation, rather than a single F-score). Additionally the method effectively deals with overfitting within the dataset.

The selection process for ensuring no repeated images (lines 181-182) is not detailed. More details required for the "trained expert scientists" required (line 190-191). The techniques mentioned in 286 need to be specified.

Validity of the findings

The authors claim that their findings can "evaluate the capacity of the model to generalize across varying environmental conditions" (line 197) is overstated, given that they only test with 2 sites and no sub divisions within those sites (that are reported in the paper).

The results that show the improvement using the data augmentation are useful, but they need to show the absolute value as well as the improvement (Table 1 and 306-315).

The improvement of the precision of the models at the expense of recall should be discussed by the authors. Do we want a model that is sure that the objects it detects are correct or do we want a model that detects as much as possible even it makes mistakes? There is no definitive answer but this is an important issue to raise when researchers are attempting to operationalise such methods.

Reviewer 2 ·

Basic reporting

This paper proposed a pipeline consisting of deep learning models for object detection tasks from images of two species. Three CNN models, YOLO7, YOLO8, and R-CNN were trained on google co-lab and evaluated via metrics such as precision, recall, and F1 scores. Among them, YOLO8 achieved the best results among those tasks. Consequently, findings such as the density is higher in one area than another from the output of those models. Overall, it's an interesting application of CV models in ecology, and the pipeline is pretty novel in the field.

Experimental design

1. Since there are only 300 images in total, cross-validation could be a better choice here.

Validity of the findings

1. Numbers are not clearly presented in Table 1.

2. It's not surprising that YOLO8 model outperformed other models since YOLO8 is more complicated. To improve the results, transformer-based models like ViT and CLIP could be considered in further research.

---

## Round 0.2 · accepted · Accept

Dear Dr. Prado and co-authors,

Your revised submission was sent to the reviewer who raised most of the points you addressed in your rebuttal letter, but we have not heard from him since. I have reviewed your responses and your revised manuscript from an editorial perspective and have concluded that it is thorough and of sufficient quality. In view of this and the positive response from the other reviewer, I am in favour of accepting this revised version of your manuscript, which I consider ready for publication.

Congratulations and best regards,

Academic Editor